# Progress in Spin Logic Devices Based on Domain-Wall Motion

**DOI:** 10.3390/mi15060696

**Published:** 2024-05-24

**Authors:** Bob Bert Vermeulen, Bart Sorée, Sebastien Couet, Kristiaan Temst, Van Dai Nguyen

**Affiliations:** 1Interuniversity Microelectronics Center (IMEC), Kapeldreef 75, 3001 Leuven, Belgium; bart.soree@imec.be (B.S.); sebastien.couet@imec.be (S.C.); kristiaan.temst@kuleuven.be (K.T.); 2Department of Physics and Astronomy, Quantum Solid-State Physics (QSP) Division, Katholieke Universiteit Leuven, Celestijnenlaan 200D Box 2414, 3001 Leuven, Belgium; 3Department of Electrical Engineering, ESAT-INSYS Division, Katholieke Universiteit Leuven, Kasteelpark Arenberg 10, 3001 Leuven, Belgium; 4Department of Physics, Universiteit Antwerpen, Groenenborgerlaan 171, 2020 Antwerp, Belgium

**Keywords:** magnetic domain wall, spin-transfer torque, spin-orbit torque, logic, magnetic tunnel junction, Dzyaloshinskii–Moriya interaction

## Abstract

Spintronics, utilizing both the charge and spin of electrons, benefits from the nonvolatility, low switching energy, and collective behavior of magnetization. These properties allow the development of magnetoresistive random access memories, with magnetic tunnel junctions (MTJs) playing a central role. Various spin logic concepts are also extensively explored. Among these, spin logic devices based on the motion of magnetic domain walls (DWs) enable the implementation of compact and energy-efficient logic circuits. In these devices, DW motion within a magnetic track enables spin information processing, while MTJs at the input and output serve as electrical writing and reading elements. DW logic holds promise for simplifying logic circuit complexity by performing multiple functions within a single device. Nevertheless, the demonstration of DW logic circuits with electrical writing and reading at the nanoscale is still needed to unveil their practical application potential. In this review, we discuss material advancements for high-speed DW motion, progress in DW logic devices, groundbreaking demonstrations of current-driven DW logic, and its potential for practical applications. Additionally, we discuss alternative approaches for current-free information propagation, along with challenges and prospects for the development of DW logic.

## 1. Introduction

For decades, the information processing industry has been driven by the dimensional scaling of Complementary Metal-Oxide-Semiconductor (CMOS) technology, enhancing the speed and power efficiency of transistors. However, maintaining a constant power density while further reducing the size of the transistor presents significant challenges, primarily due to leakage currents and the large number of interconnects [1]. As CMOS scaling approaches fundamental limits, new devices and architectures are being explored to prolong the scaling trend of integrated circuits for high-speed, low-power, and high-density technologies. This includes extending the functionality of the CMOS platform via the integration of new technologies (“More Moore”) and stimulating new information processing paradigms (“Beyond CMOS”) [1].

Spintronics, utilizing both the charge and spin of electrons for computation or data storage, stands as a cornerstone in the realm of beyond-CMOS technologies. The field of spintronics was initiated in 1988 by the discovery of giant magnetoresistance (GMR) in magnetic multilayers [2,3,4]. This led to the development of read heads for hard disk drives (HDDs). Tunneling magnetoresistance (TMR) [5,6] in magnetic tunnel junctions (MTJs) made of two ferromagnetic layers separated by an oxide tunneling barrier (see Figure 1a) then enabled the development of nonvolatile magnetoresistive random access memory (MRAM). The discovery of spin-transfer torque (STT) [7,8,9], achieved by current injection through the MTJ, provided a scalable switching scheme. Advancements, including high TMR using MgO-based tunneling barriers and interfacial perpendicular magnetic anisotropy (PMA) [10], enhanced the scalability, speed, and efficiency of MRAM. STT-MRAM is now commercialized as a viable replacement for embedded flash (eFlash) memory or static RAM (SRAM) in embedded cache memory [11]. Furthermore, ongoing research is exploring alternative switching schemes to further enhance MRAM performance. A promising avenue is spin-orbit torque (SOT) [12,13], where torque is induced by passing an electrical current along a heavy metal layer in contact with the free layer. This approach aims to achieve faster and more reliable writing operations by leveraging separate writing and reading paths.

Considering that the energy to switch a nanomagnet is several orders of magnitude lower than the energy required for computation in conventional CMOS devices, there is also significant interest in using spin for ultra-low-power and/or nonvolatile logic circuits [11]. Various approaches are being investigated, including nanomagnet logic (NML) [14,15,16], magnonic or spin-wave logic [17,18], spin field-effect transistors [19], spin-based semiconductor logic [20], magneto-electric spin-orbit (MESO) logic [21], ferroelectric spin-orbit (FESO) logic [22], exchange-driven magnetic logic [23], domain-wall (DW) logic [24,25] and skyrmion logic [26]. However, in contrast to MRAM, spin logic concepts, where spin is exploited to simultaneously store and process information within a logic circuit, are still mainly in the exploration stage.

Among the different spin logic concepts, DW logic holds significant potential by exploiting the motion and interactions of magnetic DWs in nanowires to build logic circuits. A magnetic DW is the boundary between two magnetic domains with opposite magnetization directions. A basic building block of DW logic, as illustrated in Figure 1, typically requires three operations: the nucleation of DWs at the inputs (writing operation), the motion of DWs in the circuit (logic operation), and the detection of DWs at the outputs (reading operation). A long history of material research on fast and efficient DW motion in nanowires has been driven by the application potential of racetrack memory. This concept was first proposed by S.S.P. Parkin [27], and the first proof of concept was demonstrated in 2008 [28,29]. In racetrack memory, bit information is stored in magnetic domains and moved by short current pulses in a nanotrack. By creating 3D magnetic tracks, it has the potential to become an ultra-high-density memory [30,31]. This has been an important driver for research on DW motion, from which DW logic directly benefits. Moreover, the writing and reading of DWs can also take advantage of the developments of MRAM. Importantly, by integrating multiple functions into a single device, such as nonvolatile majority gates, DW logic enables functional scaling. This capability holds promise for building compact logic gates and simplifying logic circuit complexity [1]. The nonvolatility also allows combining logic and memory for in-memory computing architectures [32]. Furthermore, the stochastic behavior of DWs can be exploited for unconventional computation paradigms, including neuromorphic computing, stochastic computing, and reservoir computing [33]. Nevertheless, the demonstration of DW logic circuits with electrical writing and reading at the nanoscale is still needed to unveil their real application potential.

This review discusses the recent progress in spin logic devices based on DW motion. We cover the progress of material research aimed at electrically driving DWs at high speeds and describe the advancements in DW logic devices with the groundbreaking demonstration of current-driven DW logic. We then delve into the integration of MTJs for complete electrical control of DW devices at the nanoscale. Furthermore, we describe alternative approaches that do not require electrical current for spin information propagation. Finally, we discuss challenges and prospects for the future development of DW logic.

## 2. Physics and Materials Research for Fast Current-Driven DW Motion

DWs can naturally be driven by the application of an external magnetic field, inducing the expansion of magnetic domains with magnetization parallel to the field and the contraction of antiparallel domains. While field-driven DW motion is crucial in material research, the required coils limit its scalability. Moreover, two adjacent DWs move in opposite directions due to the expansion or contraction of the domain in between, while typical applications require all DWs to move in the same direction.

In contrast, current-driven DW motion holds considerable technological potential in nanoscale devices by offering directional control of DW motion based on the current polarity. Research efforts have focused on mechanisms and materials to enable efficient and reliable DW motion and are summarized in Table 1. By injecting an electrical current into a ferromagnet (FM) with a DW, the current becomes spin-polarized, and spin-transfer torque (STT) induces the motion of the DW in the direction of the electron flow [34]. STT can drive DWs in FMs with in-plane (IP) magnetic domains, as well as with out-of-plane (OOP) domains resulting from perpendicular magnetic anisotropy (PMA). Faster and more efficient DW motion can be obtained by sending a current in a heavy metal (HM) with strong spin-orbit coupling (SOC) in contact with a thin FM, leading to spin-orbit torque (SOT) [34]. Finally, systems with antiferromagnetic (AF) coupling have been demonstrated to offer ultra-fast and efficient DW motion due to an additional exchange coupling torque (ECT) [35]. This can be achieved in synthetic antiferromagnets (SAF) or ferrimagnets (FiM). In the following, we review these different driving schemes and discuss the performance reached in recent years.

### 2.1. Spin-Transfer Torque

Driving DWs in an FM with an electrical current was initially proposed by Berger in 1978 [36], followed by experimental evidence in permalloy (NiFe) [37,38,39], (see inset (a) in Table 1). This research topic has been reviewed theoretically [40,41,42] and experimentally [43,44].

The DW motion by STT can be divided into an adiabatic and a nonadiabatic torque. Adiabatic torque results from spin angular momentum transfer between the spin-polarized current and the DW magnetization, leading to DW motion in the direction of the electron flow for currents above a certain threshold (typically $10^{13}$ and $10^{12}$ A/m^2^ for IP and OOP magnetized systems, respectively) [41,43,45]. For lower current densities, DWs can be driven by nonadiabatic torque, which is equivalent to a field along the easy axis [46].

**Table 1 micromachines-15-00696-t001:** Advances in current-driven domain-wall motion, with the corresponding velocity (*v*) and current density (*j*), as well as the advantages (Adv) and limitations (Lim) for practical applications. (*) indicates that the DW velocity was obtained with the application of the IP field. Based on [34].

	System	Material	v (m/s)	j ($10^{12}$ A/m^2^)	Year [Ref.]	Adv (+) / Lim (−)
Spin-transfer torque (STT) 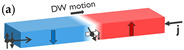	IP FM PMA FM	NiFe Co/Ni multilayer	110 40	1.5 1.4	2007 [47] 2008 [48]	(+) Simple implementation (−) Slow (−) High current density
Spin-orbit torque (SOT) 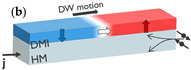	PMA FM	Pt+Co Pt+Co/Ni/Co	400 200	3.2 2.5	2011 [49] 2013 [50]	(+) Fast (+) Efficient
Exchange-coupling torque (ECT) 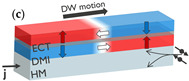 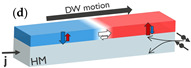	PMA SAF	Pt+Co/Ni/Co/ Ru/Co/Ni/Co (SOT)	750	3.0	2015 [51]	(+) Very fast (+) Very efficient (+) High density (+) Robust against external field
	PMA FiM	Mn_4−x_Ni_x_N (STT) Pt+CoGd (SOT) *	3000 5700	1.2 0.42	2021 [52] 2020 [53,54]	(+) Ultra-fast (+) Ultra-efficient (−) Highly temperature sensitive

STT-driven DW motion was initially studied in IP-magnetized systems. Hayashi et al. obtained a DW velocity of 110 m/s at a current density of $1.5\times 10^{12}$ A/m^2^ in 300 nm wide permalloy nanowires [28,47]. Interface-induced PMA in thin films was predicted to increase STT-driven DW motion efficiency, lower the threshold current [55], improve thermal stability, and allow for higher DW density due to the decreased DW width [34]. STT-driven DW motion was shown in PMA systems such as Co/Ni [45,48,56,57,58] and Co/Pt [59,60] multilayers, achieving velocities up to 100 m/s, as well as in GaAs, MnAs [61], and CoFeB/MgO [62,63,64] structures.

The discovery of current-driven DW motion initiated a new field of spintronics and significantly broadened the application potential. A critical technological hurdle lies in reducing the current density for DW motion, typically on the order of $10^{12}$ A/m^2^, to mitigate Joule heating and power consumption.

### 2.2. Spin-Orbit Torque

An important advancement in current-driven DW motion research was achieved through the groundbreaking studies by Moore et al. [65] and Miron et al. [49,66]. They found that the DW motion direction is opposite to the electron flow in a Pt/Co/AlO_x_ stack, in contrast to what is expected with conventional STT [67]. This initiated research on SOT-driven DW motion in FM thin films on an HM, leading to more energy-efficient and faster DW motion (see inset (b) in Table 1).

SOT-driven DW motion requires spin-orbit coupling (SOC), structural inversion asymmetry (SIA), and DWs with a Néel component [34,46]. Both the Rashba–Edelstein effect (REE) and the spin Hall effect (SHE) can contribute to the origin of SOT [46]. When injecting a charge current, the REE is an interfacial effect polarizing the spins of the electrons, leading to a spin accumulation that can exert a torque on the magnetic film. On the other hand, the SHE is a bulk effect in materials with strong SOC, leading to spin-dependent scattering where opposite spins scatter in opposite directions. A spin current orthogonal to the charge current arises, exerting a torque on the magnetization. The magnitude of the SHE is quantified by the dimensionless spin Hall angle (SHA), defined as the ratio between the spin current and the injected charge current. HMs such as Pt, Ta, and W are materials with high SHA values around 0.07, 0.15, and 0.5, respectively [46]. The SOT, generated by REE or SHE, can be divided into damping-like (DL) and field-like (FL) torques, leading to DW motion. The DL torque rotates the magnetization toward the spin current polarization, while the FL torque leads to the precession of the magnetization around the spin current polarization [46].

Interestingly, Haazen et al. showed that an in-plane (IP) field was required to convert Bloch DWs into Néel DWs to obtain SOT-driven DW motion in a Pt/Co/Pt stack [68]. However, by applying an IP field, adjacent DWs have opposite chirality (or handedness) and move in opposite directions. Alternatively, the Dzyaloshinskii–Moriya interaction (DMI) [69,70], an antisymmetric exchange interaction arising at the HM/FM interface, can stabilize Néel DWs with fixed chirality without an external field [30]. This allows adjacent DWs to move in the same direction [50,71]. By stabilizing the Néel DW configuration with DMI, faster SOT-driven DW motion is achieved with velocities reaching 400 m/s at $3\times 10^{12}$ A/m^2^ in Pt/Co/AlO_x_ [49] or Pt/Co/Ni/Co [50,72]. Slightly lower DW velocities are obtained with Ir or Pd instead of Pt due to a lower SHA [73].

SOT-driven DW motion is a significant progress compared to conventional STT. It allows for more efficient and faster manipulation of DWs, promising enhanced performance in DW devices. However, achieving the full potential of SOT-driven DW motion requires ensuring reliable DW motion in nanoscale devices. Indeed, DW tilting can lead to asymmetric device behavior [74]. Moreover, various factors, such as interface roughness and edge roughness induced during nanofabrication, can impede DW motion due to DW pinning [46].

### 2.3. Exchange-Coupling Torque

Another great improvement was achieved using AF-coupled structures. In an SAF, two magnetic layers are AF coupled through a spacer layer such as Ru [75] (see inset (c) in Table 1). The AF exchange field results in an exchange coupling torque (ECT), which is maximized when the magnetization of the two magnetic layers is equal and leads to increased DW mobility [35]. DW velocities above 750 m/s have been reported at $3\times 10^{12}$ A/m^2^ in a Co/Ni/Co/Ru/Co/Ni/Co SAF structure deposited on Pt [51]. Moreover, the SAF configuration reduces DW tilting, makes the DWs less sensitive to external magnetic fields, and suppresses stray fields that could lead to dipolar interactions between DWs within and between nanowires [34].

Besides SAF structures, ferrimagnetic alloys or multilayers composed of rare-earth (RE) metals and transition metals (TMs) exhibit AF coupling between the magnetic moments of the RE and TM materials (see inset (d) in Table 1) [34]. Since the AF coupling does not require a spacer layer, the exchange coupling between these elements can be stronger than in SAF structures. The DW mobility is maximized when the angular momentum of both sublattices is compensated, either by varying the temperature or composition [54,76,77,78,79,80]. DW velocities of 3 km/s were observed in Ni-doped Mn_4_N using STT [52] and 5 km/s in Pt/CoGd using SOT [53]. Since the angular momentum compensation point depends on temperature, these dynamics are highly temperature-sensitive, and Joule heating can significantly affect DW motion. This extreme temperature dependence currently makes ferrimagnets less attractive than SAF for practical applications.

Finally, antiferromagnets (AFMs) are also expected to allow fast current-driven DW motion [81]. Recently, significant advancements have been made in the manipulation and detection of AFM domains [82,83]. Nevertheless, due to the complete compensation of the magnetization of both sublattices, the detection of AFM domains remains a major challenge impeding their implementation in DW devices.

### 2.4. Conclusions

The discovery of current-driven DW motion has opened a new field of spintronics for both fundamental and technological applications. The development of SOT-driven DW motion in a PMA ferromagnet, with DMI to stabilize Néel DWs, provides scalable, fast, and efficient DW motion. The additional ECT within SAF structures allows for faster and more efficient DW motion and provides stability against external magnetic fields. While ferrimagnets can enable even higher DW velocities, their strong temperature dependence makes them less appealing for practical applications and more challenging to integrate with CMOS. Therefore, SOT-driven DW motion in a ferromagnet or SAF is the most promising route toward practical DW logic applications.

These findings have allowed identifying key materials to enable high-performance DW devices. Among these, Pt/Co-based stacks emerge as strategic materials for practical DW applications thanks to their high PMA and DMI, facilitating fast SOT-driven DW motion. Moreover, Co can be used either as a single layer or in various configurations such as multilayers coupled with Ni, in an SAF structure, or in a ferrimagnet with Gd.

Further challenges must be overcome to improve the potential of current-driven DW motion. Lower driving currents are required for improved power efficiency, and issues of DW pinning and thermal stability must be addressed. Finally, for practical applications, the motion of DWs must be controllable (for example, by using artificial pinning sites [46]), reliable, and reproducible.

Other schemes, including electric fields, laser pulses, or strain, can be used to drive DWs [46]. However, current-driven DW motion is the most conventional for applications as it is easily implemented and compatible with the CMOS fabrication process.

In the next section, we discuss the logic functionalities that emerge from the properties of DWs.

## 3. Emerging Logic Functionalities in DW Devices

The ability to drive DW motion by a field or by an electrical current in nanowires opens a novel paradigm for transporting spin information in logic circuits. The interactions of DWs at nanowire junctions can then be exploited to build logic gates [24,33]. Therefore, the controlled DW evolution in a circuit allows Boolean logic where information is encoded in the magnetization direction or the chirality of DWs. Efficient current-driven DW motion combined with the nonvolatility lends itself to low-power computing applications. Moreover, the collective behavior of magnetization allows for the implementation of compact logic gates, such as majority or minority gates, providing more functionality in single devices [1].

While the stochasticity of DW behavior, especially in scaled devices where thermally driven processes become significant, can be detrimental for Boolean logic applications, it naturally fits unconventional computing applications [33]. However, experimental demonstrations are still in the very early stages.

In this section, we first present the experimental advancements of DW devices for logic applications. We then describe the resulting demonstration of current-driven DW logic circuits and discuss its potential and the related challenges.

### 3.1. Advancements of DW Devices for Boolean Logic and Unconventional Computing

A list of experimental advancements of DW devices for Boolean logic functionalities is presented in Table 2. The first Boolean logic functionalities using field-driven DW motion were demonstrated by Allwood et al. [24]. They used a rotating IP field to drive DWs in a circuit of IP-magnetized nanowires, where the information is encoded in the direction of the magnetization. The DW behavior at the nanowire junctions allows for the implementation of logic functions, including NOT, AND, NAND, OR, and NOR gates [24,84,85]. Further work with IP nanowire elements led to shift register and transistor functionalities [86]. Instead of encoding the information in the magnetization direction, Goolaup et al. and Omari et al. suggested encoding it in the chirality of transverse DWs (TDWs) or vortex DWs (VDWs) [87,88]. Most studies used magnetic imaging to characterize the logic functionalities and read the information. Interestingly, Raymenants et al. demonstrated electrical writing and reading of DWs using STT and TMR with three MTJs on a DW track combined with field-driven DW motion. In addition to providing a scalable and efficient way to control inputs and outputs, they demonstrated the “2x + 1” arithmetic function and the shift register operation [89].

A great advancement toward practical applications is the discovery and development of current-driven DW motion, as described in Section 2. It indeed provides a local, scalable, efficient, easily implemented, and CMOS-compatible way to control DWs. Incorvia et al. demonstrated a DW-MTJ, where a DW was moved using STT in an IP-magnetized track, affecting the TMR of an MTJ on top of the track [90]. They showed that it can be used for buffer and inverter operations and later extended the device concept with SOT-DW motion in a PMA track [91]. A breakthrough was the demonstration of current-driven DW logic circuits by Luo et al. [25]. DWs were driven using SOT in µm-scale magnetic tracks with PMA, and the inverter operation was enabled by patterned IP-magnetized regions inside the DW tracks. They showed reconfigurable NAND/NOR gates as well as a one-bit full-adder circuit and diode operation [25,92]. This demonstration and the related challenges are described in more detail in Section 3.2. Importantly, Manfrini and Raymenants et al. provided a way to extend the DW logic concept toward the nanoscale with electrical writing and reading of DWs [93,94]. They indeed demonstrated electrical writing using STT and reading using TMR with MTJs, combined with DW transport using SOT in nanoscale devices. The device concept and its implications are discussed in more detail in Section 4. Finally, Zhang et al. also proposed logic gates based on optoelectronic DW motion, which combines optical laser pulses with SOT, thereby opening new avenues to control DW motion and perform logic operations [95].

DW motion also has great potential for unconventional computing applications. More information can be found in the review by Venkat et al. [33]. DW devices can indeed be used to implement neuromorphic computing (NC), mainly by providing synaptic behavior with multilevel resistance [96,97,98,99,100,101,102] or by providing a sigmoid passing probability [103,104]. DW devices also have applications for stochastic computing (SC) [105,106], as well as reservoir computing (RC) [107,108]. However, experimental demonstrations are still in the early stages. Larger networks and more advanced functionalities need to be demonstrated, assessing issues of reproducibility and robustness, and identifying applications that are suited for DW devices.

In conclusion, multiple experimental works over the past two decades have proven the potential of DW devices for both Boolean and unconventional computing. Demonstrations of compact DW logic circuits, combined with electrical writing and reading using MTJs, are promising for practical applications. The following section describes in more detail the concept of current-driven DW logic circuits and the related challenges.

### 3.2. Current-Driven DW Logic Circuits

In 2019, Luo et al. demonstrated that adjacent magnetic regions with PMA, separated by an IP-magnetized region, can be chirally coupled due to the DMI [109]. The DMI is an antisymmetric exchange interaction [69,70], with Hamiltonian $H_{DMI}=-D_{ij}\cdot \left(S_{i}\times S_{j}\right)$ that favors orthogonal alignment of neighboring magnetic moments $S_{i}$ and $S_{j}$ with a fixed chirality given by the DMI vector $D_{ij}$. This fixed chirality leads to antiparallel alignment of the OOP regions separated by an IP region. They demonstrated chiral coupling in a Pt/Co/Al stack with strong DMI coupling originating from the Pt/Co interface and with a tunable PMA depending on the oxidation degree of the Co/Al interface (see Figure 2a).

The discovery of chiral coupling between two PMA regions separated by an IP region led to the demonstration of an SOT-driven DW inverter (i.e., NOT gate) [25], a fundamental building block of any Boolean logic circuit. The DW inverter operation was enabled by a narrow IP region inside the PMA DW tracks. As the two OOP regions on either side of the IP region were coupled by the DMI, the reversal of one OOP region induced the reversal of the other, leading to the inversion of a DW traveling along the track (see Figure 2b,c). Note that the IP regions were V-shaped to facilitate the DW nucleation.

More complex logic gates with multiple inputs could then be fabricated based on the inverter. Inputs were set by saturating with an OOP external magnetic field and placing inverters at specific locations. A reconfigurable NAND/NOR gate was made with three input tracks, a U-shaped IP region, and one output track (see Figure 2d). The multiple magnetic tracks were patterned on one large Pt track for SOT. This gate corresponds to a three-input minority gate, where the value of the output follows the minority of the three inputs. Importantly, the inverter and the minority gate can be used to build any Boolean logic circuit. For example, they also showed more complex circuits by combining NAND gates to fabricate an XOR circuit or a one-bit full adder (see Figure 2e).

### 3.3. Conclusions

The demonstration of SOT-driven DW logic circuits is a significant step forward, showing the potential of DW motion for logic circuit implementation. Nevertheless, some challenges need to be addressed for real industrial applications. The use of a wide common SOT track for all the magnetic tracks implies that only a fraction of the current (i.e., the current flowing underneath the magnetic tracks) is effectively used for SOT-driven DW motion. Moreover, issues of DW velocities in different arms, synchronization, and current distribution need to be addressed. Most importantly, the concept needs to be scaled down, requiring electrical writing and reading at the nanoscale instead of using an external magnetic field and magnetic imaging. This would allow for the demonstration of more complex logic circuits and networks to be tested for repeatability, reliability, and precise manipulation of DWs at the nanoscale and high speeds.

In the next section, to address these missing pieces, we describe how MTJs are a CMOS-compatible option to write (nucleate) and read (detect) DWs in a nanowire, paving the way to reach the full potential of DW logic circuits.

## 4. DW Devices with MTJ Write and Read at the Nanoscale

The demonstration of current-driven DW logic by Luo et al. [25] is a great advancement for spintronic logic. However, the main hurdle limiting the practical application of this concept is the lack of electrical writing and reading of the DWs at the inputs and outputs, preventing scaling toward the nanoscale and integration with CMOS devices. In this section, we first discuss several techniques to electrically write and read DWs in a nanowire. We then present a hybrid free layer stack design enabling the integration of MTJs in DW devices and describe the experimental demonstration in nanoscale devices.

### 4.1. Electrical Writing and Reading of DWs

Various techniques can be employed to electrically detect DWs in FM materials [110]. DW measurements at the nanoscale are often performed in Hall bar devices, in which the Hall resistivity depends on the magnetization due to the anomalous Hall effect (AHE) [111]. While it enables the determination of the DW position, the readout signal is low, and the scalability is limited because of the required four contacts. Anisotropic magnetoresistance (AMR) allows for the detection of a DW in an IP-magnetized wire between two contacts but not the exact DW position [39]. Magnon magnetoresistance (MMR) enables the determination of the DW position in a PMA or IP-magnetized wire but also suffers from a poor readout signal [112,113]. In contrast, GMR provides a localized detection of the DW with a high readout using spin valves [4]. TMR provides an even higher readout signal in MTJs formed by a fixed and a free magnetic layer separated by an oxide spacer [5,6] (see Figure 1a). Moreover, MTJs are scalable and compatible with current CMOS processing. Therefore, TMR in MTJs is a promising solution to electrically read DWs in nanoscale devices.

The different MRAM writing schemes can be considered to nucleate DWs in nanotracks with PMA [34,110]. Nucleation pads are often used to nucleate DWs by applying an external magnetic field. Although easily implementable for research, the use of an external magnetic field is neither scalable nor energy-efficient in device applications. DWs can also be nucleated by the magnetic Oersted field induced by an electrical current in wires. However, the large currents required to generate a sufficient Oersted field also limit scalability. In contrast, MTJs with STT offer a fast and scalable method to nucleate DWs by applying an electrical current vertically through the MTJ. The main limitation is the high current densities required to provide sufficient torque, which can lead to the breakdown of the oxide tunneling barrier and limit the endurance. An alternative is to send a current in an HM in contact with the magnetic layer to write DWs using SOT. Although less compact than the STT writing scheme, this provides improved endurance and promises sub-ns switching times with sub-pJ energies. To further decrease the writing energy, voltage-controlled magnetic anisotropy (VCMA)-MRAM uses a gate voltage to reduce the anisotropy of the free layer and induce its switching. However, the current magnitude of the VCMA effect only enables the reversal of free layers with low retention. Finally, all-optical schemes for ultra-fast switching have been demonstrated, but their practical implementation is still under investigation [114,115,116]. Other schemes include the patterning of an IP region for DW injection using STT or SOT [117] and thermally assisted DW nucleation [118].

While SOT and VCMA switching typically require an assisting IP field, STT is a field-free and compact solution to locally induce DW nucleation. As a result, MTJs with STT write and TMR read provide a practical and scalable approach. Moreover, the same device can then be used as input and output for more flexibility.

Even though MTJ pillars are excellent candidates for writing and reading units in DW devices, their integration presents material and fabrication challenges. First, MTJs optimized in STT-MRAM are based on a CoFeB free layer and a MgO tunneling barrier, providing high STT efficiency and TMR readout. Conversely, materials researched for fast and efficient DW motion are typically based on Co and Pt. Therefore, it is challenging to obtain materials providing both good MTJ properties and DW motion efficiency. Second, integrating MTJ pillars on a DW track requires etching the pillars down to the tunneling barrier to avoid any electrical short between input and output MTJs while preserving an intact DW track in between the MTJs. To address these challenges, Raymenants et al. developed a hybrid free layer (HFL) stack design combining the conventional free layer from the MTJ stack with a second free layer for DW motion [94]. Such a design had been studied to increase the thermal stability of STT-MRAM memory cells [119,120,121] but had not yet been explored for DW devices.

### 4.2. Hybrid Free Layer Concept

In the HFL stack design depicted in Figure 3a, the first free layer (FL1) consists of a conventional CoFeB/MgO-based free layer, providing efficient STT and high TMR. The second free layer (FL2) typically consists of an FM or an SAF deposited onto a heavy metal and serves as a DW conduit to efficiently transport DWs. FL2 offers high DW speed originating from interfacial DMI, SOT, and/or ECT. The two free layers are FM coupled via interlayer exchange coupling through a spacer, such that they behave as a single free layer. The spacer also plays a crucial role in decoupling the crystallization of CoFeB (FL1) and FL2, as the face-centered cubic (111) crystal structure of Co is incompatible with the body-centered cubic (001) crystal structure of CoFeB [94]. This also allows the growth of the MgO tunneling barrier for retaining high TMR and high STT efficiency. This approach provides flexibility to integrate a variety of materials as FL2 depending on the application.

Figure 3b,c show that this stack allows for tuning the MTJ properties with the CoFeB thickness while providing flexibility for FL2 for DW motion materials. Indeed, in Figure 3b, TMR mainly depends on the CoFeB thickness and reaches high values for CoFeB above 8 Å. This allows choosing a CoFeB thickness that provides a great balance between a high readout signal and high PMA [122]. Moreover, Figure 3c shows that different DW conduit materials can be used as FL2, such as Pt/Co or an SAF, without greatly affecting the TMR of the MTJ.

Figure 4a illustrates the device concept. STT is locally applied at the input pillar to inject DWs into the DW conduit via the strong FM coupling between CoFeB and FL2. Subsequently, the DW travels along the track driven by a magnetic field or SOT. The detection of the DW at the output is enabled using TMR.

Finally, this stack provides the required robustness against the pillar patterning step [123]. For STT-MRAM, the MTJ pillars can be etched down to the seed layer. In contrast, for DW devices, the pillar patterning requires a controlled stop on the MgO layer to avoid any electrical shorts between the pillars, while the DW conduit materials need to stay intact with good PMA. Ion beam etching (IBE) is a standard technique in nanofabrication, but the ion penetration depth of a few nm makes it challenging to precisely etch down to the MgO while maintaining the properties of the DW track underneath. As shown in Figure 4b, with the HFL stack, even if the PMA of the CoFeB is affected by the etching, the deeper DW conduit can maintain good PMA.

### 4.3. Electrical Operation of Nanoscale DW Devices with MTJ Write and Read

Figure 5 demonstrates the device operation at the nanoscale with MTJ writing and reading, fabricated on imec’s 300 mm wafer platform. The device comprises three MTJ pillars (P1, P2, and P3) on a common DW track. The track is initialized in the “down” state using an external magnetic field larger than the coercive field. STT is then applied to locally write an “up” domain in P2. In Figure 5a,d, the “up” domain is expanded with a magnetic field much smaller than the coercive field such that it can propagate DWs but not nucleate a new domain. This leads to a symmetric expansion of the domain toward P1 and P3. In Figure 5b,c,e,f, the DW is driven by an SOT current instead of a magnetic field. Depending on the direction of the SOT current, the domain propagates to P1 or P3.

### 4.4. Potential and Challenges of Electrically Controlled DW Logic

The combination of the works of Luo et al. [25] and Raymenants et al. [94] holds great potential for DW logic circuits with electrical writing and reading using MTJs, as depicted in Figure 1. Luo et al. demonstrated current-driven DW logic where the inverter gate consists of an IP magnetic region inside a PMA DW track. It was patterned by selectively oxidizing a Pt/Co/Al stack, converting it from IP to PMA. On the other hand, Raymenants et al. developed an HFL stack with PMA combining an MTJ stack with DW materials. This stack could be used for DW logic, provided that a treatment is used to locally convert some regions from PMA to IP for the logic gates. Techniques including IBE or ion irradiation could be considered to control the PMA in the DW tracks [122].

One major challenge for the practical implementation of current-driven DW logic is the efficient management of the driving current. Indeed, in [25], the magnetic tracks were patterned on one common SOT track based on Pt. This provides a uniform current density along the circuit. However, it implies that some fraction of the current does not contribute to SOT where the Pt is not covered by magnetic tracks. Moreover, it leads to different DW velocities depending on the orientation of the tracks, which can lead to DW synchronization issues. Alternatively, the SOT track could also be patterned simultaneously with the magnetic tracks such that the current only flows under the magnetic tracks. This would prevent any unused current and allow DW motion in any direction with the same velocity. However, this would lead to a complex current distribution, as depicted in Figure 6, especially where magnetic tracks merge and split, which could alter DW motion.

For these reasons, several concepts have been proposed to propagate and process information between input and output MTJs without requiring any SOT current. Those concepts are discussed in the next section.

## 5. Current-Free Alternatives to SOT DW Logic

To avoid the challenges linked to SOT-driven information propagation listed in Section 4.4, several alternatives allow for direct parallel or antiparallel coupling between the input and the output, without requiring the application of current.

Imre et al. demonstrated nanomagnet logic using dipolar coupling in permalloy nanomagnets in 2006 [15]. Breitkreutz et al. later demonstrated a majority gate based on dipolar coupling with PMA nanomagnets, providing higher thermal stability compared to IP permalloy [16]. Due to the dipolar field, the antiparallel ordering of adjacent nanomagnets represents the lowest energy state. Hence, an alternating external clocking field with an adequate amplitude sets the nanomagnets in the antiparallel state. A majority or minority gate can be constructed by coupling three input nanomagnets to one output nanomagnet. By using the output of each gate as input for consecutive gates, the authors were able to build more complex logic circuits where information is driven by the clocking field. While this is an elegant way of performing nanomagnetic logic, its potential is limited due to the required external clocking field. Moreover, the inputs and outputs are not electrically controlled, and using MTJs raises concerns about the potential interference with the stray field from the MTJs.

In 2011, Nikonov et al. proposed a more scalable concept in which input and output MTJs share a common free layer, providing direct coupling via the exchange interaction [125]. By placing the MTJs at a distance comparable to or smaller than the DW width, the equilibrium state of the free layer has uniform magnetization, favoring parallel alignment of the input and output MTJs. By coupling four MTJs via a cross-shaped free layer, the authors showed that a spin-torque majority gate (STMG) can be implemented, where inputs are switched using STT and the output is read using TMR. While this concept does not require any external clocking field and includes electrical control of the inputs and outputs, exchange coupling is only effective over distances comparable to the DW width, typically 10 to 20 nm in PMA magnets [126,127]. Therefore, this concept is only applicable to ultra-scaled dimensions, which are beyond current fabrication capabilities and can lead to thermal stability issues, preventing its experimental demonstration. Moreover, since exchange coupling leads to parallel alignment, the required inverter operation is not trivial [128].

In 2024, Vermeulen et al. proposed to couple adjacent PMA MTJs through chiral coupling using an interconnecting IP-magnetized free layer and the DMI [124]. Since the DMI favors a fixed rotation direction of the magnetization, it leads to antiparallel coupling of adjacent MTJs. While chiral coupling was experimentally demonstrated by Luo et al. in 2019 [109], Vermeulen et al. showed that by tuning the PMA and the DMI, the chiral coupling can be the driving force to switch a PMA nanomagnet. Therefore, it allows for the implementation of an inverter and a compact minority gate without requiring an external magnetic field, as depicted in Figure 7. Since the interconnection is IP, it is effective over distances comparable to the width of a TDW, typically 50 to 100 nm, alleviating the fabrication challenge of the exchange coupling concept. Moreover, the HFL can be leveraged to combine the MTJ materials for writing and reading with high DMI materials for the coupling. To control the anisotropy of the interconnecting free layer, treatments including selective oxidation, IBE, or ion irradiation can be used [122]. While this proposal provides a compact solution for electrically controlled inverters and minority gates, the main challenge toward practical implementation lies in achieving precise tuning of the PMA and the DMI to ensure operation.

Both the exchange coupling and chiral coupling concepts are scalable approaches; however, cascading information over a large distance in the magnetic domain is still under investigation. While electrical control with MTJs allows for cascading in the electric domain, the overhead due to the additional energy and footprint for cascading could outweigh the gains of performing the logic in the magnetic domain. Finally, the concept of chirally coupled MTJs provides an unconventional avenue to explore the intriguing physics of DMI and magnetic frustration in device operation through the electrical detection of MTJs.

## 6. Conclusions: Challenges and Prospects

DW logic has the potential to impact the microelectronics industry by enabling compact, low-power, and nonvolatile logic circuits. We reviewed the progress in material research for fast current-driven DW motion, the demonstration of DW logic functionalities, and the electrical control of DW devices using MTJs.

The development of SOT-driven DW motion in ferromagnets with PMA and DMI enabled fast and efficient DW motion. The addition of ECT in SAF structures further improved performance, with a low sensitivity to external perturbations, making it appealing for future applications. Further research is needed to decrease the driving currents and DW pinning for ultra-low-power DW motion. New materials such as ferrimagnets and antiferromagnets are currently being investigated but can suffer from strong temperature sensitivity and challenging detection, respectively. Importantly, conventional applications require controlled and reproducible DW motion. Artificial pinning sites are being researched to achieve more controlled DW motion while keeping low driving currents.

The demonstration of SOT-driven DW logic circuits is an important step in revealing the functionality of DWs. The logic gates are based on a DW inverter made of a narrow in-plane magnetic region inside PMA DW tracks with chiral coupling originating from the DMI. To unlock its full potential, this concept should be scaled down requiring electrical writing and reading of DWs at the nanoscale.

MTJs are excellent candidates for compact and efficient DW writing and reading units using STT and TMR, respectively, and benefit from the optimization of MRAM technology. The HFL design is an effective approach to combining DW materials with the MTJ stack, allowing electrical control in nanoscale DW devices. Further work is required to demonstrate fully electrically controlled DW logic circuits at the nanoscale. The effect of complex current distributions in such circuits also entails investigation.

Alternatively, adjacent MTJs could be chirally coupled using DMI without requiring SOT-driven DW motion, leading to compact and potentially faster and more efficient logic gates. Achieving device operation will require precise tuning of the PMA and the DMI of the material stack. Furthermore, efficient cascading schemes must be researched. Interestingly, such concepts provide new directions to explore the physics of chirality and magnetic frustration with electrical detections of MTJs.

The main requirements for the commercialization of DW-based logic include nanoscale feature size, back-end-of-line (BEOL) compatibility, and ensuring low-power and reliable operation. The current state of MTJ technology and DW devices provides a promising pathway to meet the first two requirements. The electrical writing and reading of DWs using MTJs indeed allows for the control and integration of nanoscale DW devices compatible with CMOS BEOL processes. A primary challenge for commercialization lies in ensuring reliable DW logic operation. Different factors such as current distribution, synchronization of DWs, DW pinning, and nucleations can significantly impact the reliability of DW logic devices. A demonstration of nanoscale electrically controlled DW logic is needed, including larger circuits combined with benchmarking to identify target applications. Further research on materials, processes, and devices can enable to achieve the required efficiency, scalability, and reliability. Finally, combining DW logic with CMOS in a hybrid system will allow the validation the concept.

## Figures and Tables

**Figure 1 micromachines-15-00696-f001:**
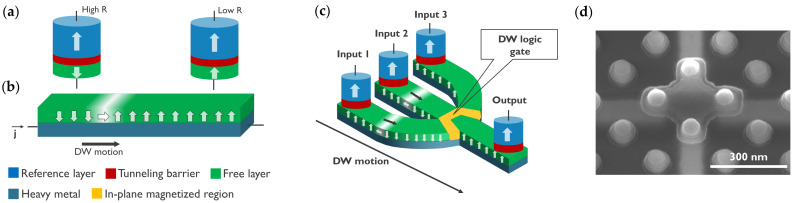
Concept of electrically controlled DW logic. (**a**) MTJ comprising a reference layer (RL) with fixed magnetization, a tunneling barrier, and a free layer (FL). The FL magnetization direction can be switched between up and down by injecting an electrical current through the MTJ due to STT. The MTJ is in a low/high resistance (R) state when the RL and FL are in a parallel and antiparallel configuration, respectively, due to TMR. MTJs can be used to nucleate (write) DWs at the inputs using STT and detect (read) DWs at the outputs using TMR. (**b**) DWs can be moved by injecting a current (j) in a heavy metal in contact with the magnetic film. (**c**) DW logic device with three inputs and one output. More complex logic circuits can be implemented with multiple logic gates and using MTJs at the inputs and outputs of the circuit. (**d**) Transmission electron microscopy image of a nanoscale DW device fabricated on a 300 mm integration platform. The DW track has a cross shape. Four MTJs are fabricated at the extremities of the cross to write or read DWs.

**Figure 2 micromachines-15-00696-f002:**
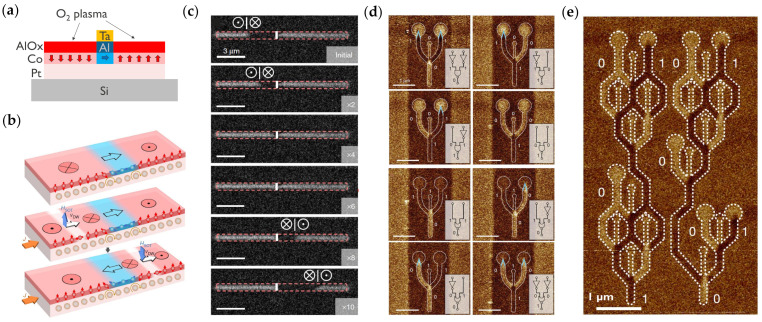
Demonstration of current-driven DW logic. (**a**) Fabrication process of PMA and IP magnetic regions. Only regions not covered by a Ta mask are oxidized by an O_2_ plasma, providing PMA. (**b**) DW inverter concept with a small IP region inside the PMA DW track with DMI. (**c**) Demonstration of the SOT-driven DW inverter by MOKE. An up-down DW is converted into a down-up DW. (**d**) Demonstration of a reconfigurable NAND/NOR gate by MFM. A NAND or an NOR gate is obtained in the case where the middle input branch is set to 0 (up magnetization) or 1 (down magnetization), respectively. (**e**) Demonstration of a one-bit full-adder circuit by MFM. Panels (**b**–**e**) are reprinted with permission from [25], Copyright 2020 by Springer Nature.

**Figure 3 micromachines-15-00696-f003:**
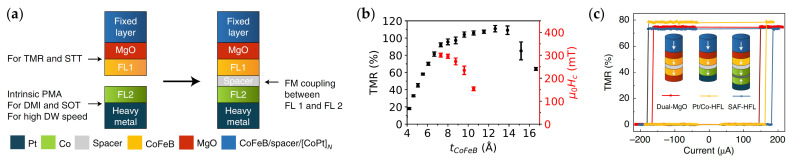
HFL concept combining MTJ and DW motion materials. (**a**) HFL consisting of FL1 based on CoFeB/MgO for high TMR read and efficient STT write, and FL2 with high DMI and PMA for fast SOT DW motion. The two FLs are FM coupled through a spacer layer. (**b**) TMR (black) and coercivity (red) from field-driven hysteresis loops, depending on the CoFeB thickness (FL1). Measurements on HFL MTJ pillars with a diameter of 100 nm. The MTJ properties can be tuned with the thickness of FL1. (**c**) TMR vs. STT current loops for 50 ns current pulses for different FL2: MgO/CoFeB, Pt/Co, and SAF. The HFL provides flexibility to introduce different materials as FL2. Panels (**a**,**c**) are reprinted with permission from [94]; Copyright 2021 by Springer Nature. Panel (**b**) is reproduced from [122]; CC BY 4.0.

**Figure 4 micromachines-15-00696-f004:**
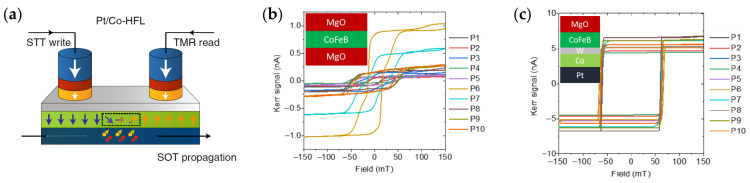
HFL concept for DW devices. (**a**) Schematic of a DW device with STT writing at the input, SOT DW motion, and TMR reading at the output. (**b**,**c**) MOKE hysteresis loops at different positions along 300 mm wafers for a dual-MgO stack and an HFL stack after ion beam etching down to the top MgO layer. In the HFL stack (**c**), the PMA is conserved and remains uniform along the wafer thanks to the Pt/Co layer. In contrast, the dual-MgO stack (**b**) has weaker and nonuniform PMA after etching. Therefore, the HFL provides improved robustness against IBE patterning. Figure reprinted with permission from [94]; Copyright 2021 by Springer Nature.

**Figure 5 micromachines-15-00696-f005:**
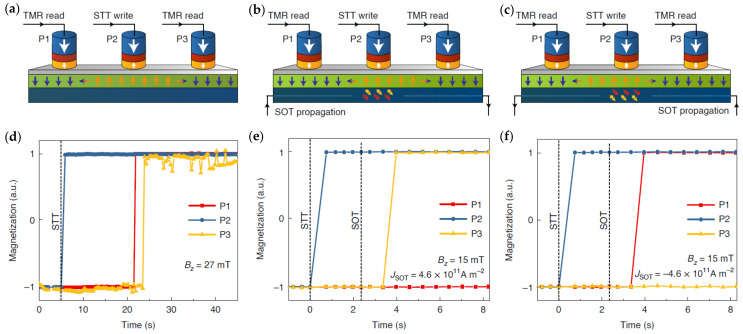
Demonstration of nanoscale DW devices with MTJ writing and reading using field-driven (**a**,**d**) and SOT-driven DW motion (**b**,**c**,**e**,**f**). DW track width of 180 nm, MTJ pillar diameter of 80 nm, and interpillar spacing of 500 nm. (**a**,**d**) Field-driven domain expansion from P2 to both P1 and P3. (**b**,**e**) SOT-driven DW motion from P2 to P3 with positive current flowing in the heavy metal. (**c**,**f**) SOT-driven DW motion from P2 to P1 with a negative current. Figure reprinted with permission from [94]; Copyright 2021 by Springer Nature.

**Figure 6 micromachines-15-00696-f006:**
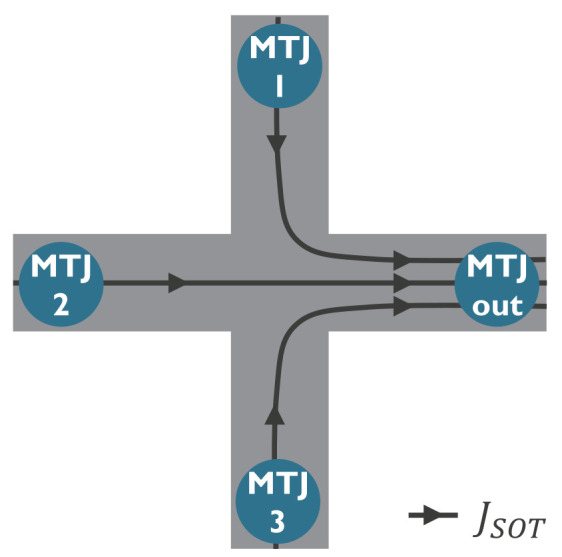
Schematic of the distribution of the SOT driving current in a DW logic device with three input MTJs and one output MTJ connected by cross-shaped DW tracks (top view). Figure reprinted with permission from [124]; Copyright 2021 by the American Physical Society.

**Figure 7 micromachines-15-00696-f007:**
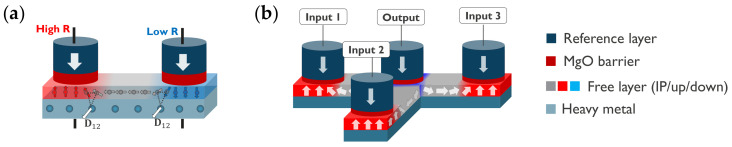
Concept of chirally coupled MTJs for logic operation without SOT current. (**a**) Inverter gate formed by two MTJs chirally coupled in antiparallel directions via an interconnecting IP free layer with DMI. (**b**) Minority gate formed by one output MTJ coupled with three input MTJs. Figure reprinted with permission from [124]; Copyright 2021 by the American Physical Society.

**Table 2 micromachines-15-00696-t002:** List of experimental advancements of DW devices for Boolean logic functionalities. The corresponding DW writing, transport, and reading schemes are indicated, as well as the DW track widths. Based on the review by Venkat et al. [33]. “Oersted” indicates that the DWs are nucleated using the magnetic Oersted field produced by the electrical current in a wire. MOKE: magneto-optic Kerr effect. MFM: magnetic force microscopy. MR: magnetoresistance. MTXM: magnetic transmission X-ray microscopy. AHE: anomalous Hall effect.

	Implementation	Functionalities	Write/Transport/Read	System	Track Width	Year [Ref.]
Field-	IP nanowire circuit	NOT, AND, NAND,	Field/Field/	IP	200 nm	2005
driven		OR, NOR, COPY,	MOKE-MFM-MR			[24,84,85,86]
		fanout, shift register,				
		transistor				
	Chirality-encoded	NOT, AND, NAND,	Field/Field/MFM-MTXM	IP	120 nm	2015
		OR, NOR				[87,88]
	Magnetically	2x + 1, shift register	STT/Field/TMR	PMA	150 nm	2018
	interconnected MTJs					[89]
Current-	Single MTJ	Buffer, inverter, fanout	Field/STT/TMR	IP	400 nm	2016 [90]
driven	on DW track	Inverter	Oersted/SOT/TMR	PMA	250 nm	2021 [91]
	PMA DW circuit	NOT, NAND, NOR,	Field/SOT/MOKE-MFM	PMA	200 nm	2020
		XOR, full adder, diode				[25,92]
	Magnetically	AND	STT/SOT/TMR	PMA	180 nm	2018
	interconnected MTJs					[93,94]
	Optoelectronic	AND, NAND, OR,	Opto-SOT/Opto-SOT/AHE	PMA	4 µm	2020
	DW motion	NOR				[95]

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
