# Peer review of "Progress in Spin Logic Devices Based on Domain-Wall Motion"

_micromachines, 2024, doi:10.3390/mi15060696_

Round 1

Reviewer 1 Report

Comments and Suggestions for Authors

This article reviews the progress in the field of spin logic devices that utilize the motion of magnetic domain walls (DWs). It explores advancements in materials that enable high-speed DW motion, discusses the integration of magnetic tunnel junctions for electrical control, and examines the challenges and potential applications of DW-based logic circuits. The review highlights how these devices offer a promising avenue for compact, energy-efficient logic circuits. In my opinion, this review is of high quality and can be published in Micromachines. Here are some minor comments for the authors to consider.

1、Spin logic devices are widely recognized as potential candidates for in-memory computing. In this paper, the efficiency, scalability, and practicality of DW-based spin logic devices are discussed. Could the author provide a comparative analysis that highlights the advantages and disadvantages of DW-based spin logic devices compared to other in-memory computing devices as well as CMOS devices?

2、Section 2 is titled “Evolution of Materials Research for Fast Current-Driven DW Motion”, but the content also discusses various DW driving schemes and physics. Could the title be optimized to better match the content discussed?

3、What does the author identify as the primary challenge currently hindering the commercialization of DW-based logic?

Reviewer 2 Report

Comments and Suggestions for Authors

In this review article, the authors summarize the experimental efforts of magnetic domain wall based logic. The review article is very well written, starting with a concise overview of spintronics and then narrowing down to discuss domain wall devices. Table 1 is very helpful for summarizing the different operating methods and main thin film layers used in domain wall devices, and Table 2 is also helpful in summarizing what logic operations have been demonstrated. The focus on experimental efforts is appreciated, since this is what is challenging in this field, rather than expanding to simulation efforts as well.

I have some minor comments for strengthening the work:

·      For Table 2, you could include the feature node (e.g. domain wall track wire width) that these functions were demonstrated at, to show where the field is at in terms of scaling.

·      While, as mentioned above, the focus on experiments is a positive thing, the authors could consider briefly summarizing larger circuit simulations and references what types of circuits domain wall devices have been simulated to properly operate in.

·      There has been some recent work on domains in antiferromagnetic and ferrimagnetic materials, consider including that as a future direction?

·      I wonder if the article could include guidance on good practices for measuring domain walls in devices, such as making sure the domains are continuous, avoiding random domain wall nucleations and how to notice those, and what should be focused on in demonstrations. For example, do domain wall devices need to be demonstrated with high TMR? Or is a detectable two-state field loop enough, since TMR can be further optimized, and research should focus more on showing the functions, cycling reliability, speed? These are just suggestions.
